# Identification of *Sarcocystis* spp. in Slaughtered Sheep from Spain and Evaluation of Bradyzoite Viability after Freezing

**DOI:** 10.3390/vetsci11030103

**Published:** 2024-02-27

**Authors:** María Paz Peris, María Jesús Gracia, Bernardino Moreno, Paula Juan-Puente, Mariano Morales, María Serrano, María Dolores Manzano, Nabil Halaihel, Juan Badiola, Juan Antonio Castillo

**Affiliations:** 1Department of Animal Pathology, Faculty of Veterinary Sciences, Universidad de Zaragoza, 50013 Zaragoza, Spain; mjgracia@unizar.es (M.J.G.); bmoreno@unizar.es (B.M.); paulajuanpuente@gmail.com (P.J.-P.); mariano@albeitar.com (M.M.); meri23sp@gmail.com (M.S.); lolismanzano@yahoo.es (M.D.M.); halaihen@yahoo.com (N.H.); badiola@unizar.es (J.B.); jacasti@unizar.es (J.A.C.); 2Instituto Universitario de Investigación Mixto Agroalimentario de Aragón (IA2), 50013 Zaragoza, Spain

**Keywords:** *Sarcocystis*, vital staining, sheep meat, trypan blue, double fluorescence staining, freezing, molecular identification

## Abstract

**Simple Summary:**

This study was designed to identify the protozoan parasite *Sarcocystis* spp. in sheep meat in Spain and to measure its inactivation when meat is frozen at −20 °C. *Sarcocystis* spp. cause substantial economic losses in the meat production sector when entire carcasses have to be condemned. One of our most significant findings is the presence of *S. medusiformis*, a species regarded as rare in Europe. Our results lead us to recommend freezing meat at −20 °C for 96 to 144 h to deactivate *Sarcocystis* spp., a procedure that results in a median viability of 1.5 to 0%, respectively. Measurements were conducted using two vital stains not previously employed for the viability assessment of bradyzoites in sheep meat.

**Abstract:**

*Sarcocystis* spp. are complex apicomplexan parasites that cause a substantial economic impact on livestock used for meat production. These parasites are present worldwide. Our study aimed to identify *Sarcocystis* species affecting sheep meat in southern–central Spain and to evaluate the effectiveness of freezing for parasite inactivation. A total of 210 condemned samples of sheep meat were thoroughly assessed grossly and microscopically; the presence of macro- and microcysts was confirmed. The samples were then frozen at −20 °C for various time intervals (24, 48, 72, 96, 120, and 144 h) and compared with untreated samples. Bradyzoites were isolated through pepsin digestion for subsequent molecular analysis and viability assessment, employing trypan blue and double fluorescence staining techniques. Our measurements confirmed the presence of *S. tenella*, *S. gigantea*, and *S. medusiformis* in Spanish domestic sheep. Freezing for 96 to 144 h resulted in a significant reduction in parasite viability, with a robust correlation observed between the two staining methods. Both stains effectively measured the viability of *Sarcocystis*, thereby promising future advances in meat safety.

## 1. Introduction

*Sarcocystis* spp. are protozoan parasites with an obligatory two-host life cycle involving sexual development in a definitive host (carnivores and omnivores) and asexual reproduction in an intermediate host (herbivores, omnivores, and carnivores). *Sarcocystis* spp. can infect a wide range of hosts, including important meat-producing animals such as cattle, sheep, and pigs, in which these parasites form mature cysts, mainly in striated muscles [1]. Domestic sheep act as intermediate hosts for four species, namely, *S. tenella* (*S. ovicanis*) and *S. arieticanis*, both of which yield microscopic cysts, and *S. gigantea* (*S. ovifelis*) and *S. medusiformis*, both of which yield macroscopic cysts. Furthermore, the species *S. microps*, *S. mihoensis*, *S. capracanis*, *S. morae*, *and S. moulei* have also been found, although relatively rarely [2,3,4].

Little is known regarding which species are present in Spain. Previous studies have identified *S. tenella* as the most common species, followed by *S. gigantea* and *S. arieticanis* [5]. *S. medusiformis* is believed to be absent [6,7]; however, a recent study by Gjerde et al. (2020) ascertained its presence in sheep in Spain [8].

According to current regulations (EU Regulation 2017/625 and Spanish Royal Decree 640/2006), carcasses affected in two or more muscular areas by *Sarcocystis* spp. macrocysts have to be condemned. This can lead to significant economic losses [2]. In a study conducted in Spain, 12% of the animals tested positive for *Sarcocystis* spp.; 79% of the carcasses were seized totally, and 21% were partially seized. This translates to an annual loss of €20,000,000 for the Spanish industry [2]. There is no ovine vaccine against *Sarcocystis* spp.; thus, control measures are the only option. The excretion of *Sarcocystis* in the faeces of the final host is an important factor in the spread of infection. It is therefore crucial to avoid feeding raw meat to cats (definitive hosts for *S. gigantea* and *S. medusiformis*) and dogs (definitive hosts for *S. arieticanis* and *S. tenella*) [7].

A method for preventing transmission to humans through the ingestion of infected meat involves the application of thermal treatment with the purpose of inactivating the parasite. Several studies have assessed the viability of bradyzoites of various *Sarcocystis* species after applying a freezing treatment in various intermediate hosts. It appears that the effect of freezing is more related to the *Sarcocystis* species than to the host species. For example, for *S. suihominis* and *S. miescheriana* in pigs and several different *Sarcocystis* species in cattle, freezing at −20 °C for 24–48 h is generally sufficient to inactivate the parasite [9,10]; some of those species cause zoonoses and therefore represent a significant concern for public health. Similar results have been observed in other species, such as several *Sarcocystis* species in guanaco (*Lama guanicoe*) [11], *S. levinei* in buffalo [12], *S. capracanis* in goat [13], and *S. fayeri* in horse [14]. However, to inactivate *S. aucheniae* in llamas, a temperature of −20 °C for 10 days is necessary [15], while only 2 h at −20 °C is required for *S. sybillensis* and *S. wapiti* in Sika deer (*Cervus nippon centralis*) meat [16]. In the case of sheep, Collins and Charleston (1980) reported that *S. gigantea* can remain viable after more than 500 days at −14 °C [17]. However, no study on the subject of parasitized sheep meat has investigated the viability of *Sarcocystis* after domestic freezing at −20 °C. 

The assessment of parasite viability is necessary in order to demonstrate the success of inactivation techniques, several of which are feasible. For example, the above-mentioned study performed by Collins and Charleston (1980) [17] assessed infectivity by feeding cats with cysts and monitoring the presence of oocysts or sporocysts in their faeces. To avoid using laboratory animals, vital stains can be a good alternative for assessing parasite viability. Trypan blue staining has been used to evaluate the viability of parasites, including *Toxoplasma gondii* [18,19,20], *Trichomonas vaginalis* [21], *Eimeria tenella* [22], *Giardia intestinalis* [23], *Schistosoma haematobium* [24], and *Taenia pisiformis* [25]. In the case of *Sarcocystis*, several studies have been conducted on *S. sybillensis* and *S. wapiti* in deer meat (*Cervus elaphus*); *S. fayeri*, *S. hominis*, *S. cruzi*, *S. hirsuta*, and *S. bovini* in horse (*Equus ferus caballus*) and beef (*Bos taurus*); and *S. sinensis* in water buffalo meat (*Bubalus bubalis*) [16,26,27]. However, no studies have been conducted on the viability of *Sarcocystis* spp. bradyzoites in sheep meat.

Another staining-for-viability evaluation is the combination of carboxyfluorescein diacetate and propidium iodide. Used in parasites, the most closely related staining technique involves applying a fluorescein derivative similar to carboxyfluorescein diacetate for the purpose of identifying *Cryptosporidium parvum* oocysts and *S. cruzi* sporocysts [28]. On the other hand, propidium iodide has been used in numerous parasite viability studies featuring flow cytometry, including studies focusing on *Eimeria tenella* [22], *Toxoplasma gondii* [29], *Taenia pisiformis* [25], *Plasmodium falciparum* [30], *Leishmania infantum* [31], and specifically, *S. neurona*, particularly in sporocysts [32]. The combination of the two dyes (double fluorescence staining) for viability assessment has been commonly applied in sperm studies and, experimentally, for evaluating *Streptococcus macedonicus* [33] as well as other lactic acid bacteria [34]. The combined use of the two dyes improves the contrast between living and dead cells; nevertheless, this combination has not been previously used for the assessment of parasite viability.

As only one previous study [6] has focused on identifying *Sarcocystis* spp. in sheep in Spain, our study aimed to identify these species in sheep meat. A second objective of our study was to determine the efficacy of temperature treatments on *Sarcocystis*-infected meat by evaluating the sensitivity of bradyzoites to freezing. Moreover, given the current need for new strategies to assess parasite viability post-treatment, we evaluated the efficacy of trypan blue and double fluorescence staining.

## 2. Materials and Methods

### 2.1. Sample Collection

Carcasses of 2-to-4-year-old adult female sheep were collected from an abattoir located in central Spain. The animals had been slaughtered according to EU Regulation 2017/625 and Spanish Royal Decree 640/2006. Slaughterhouse sources were farms located in Castilla-La Mancha, Extremadura, and Andalusia; extensive livestock production is common in all of those regions. Samples were collected between May 2019 and February 2020. Two to three fragments were taken from various muscles of each carcass, including the costal muscle, hind leg, abdomen, armpit, and foreleg. Samples were collected by an official veterinarian, stored under refrigeration, and promptly transported to the Parasitology Laboratory of the Department of Pathology, Faculty of Veterinary Medicine, University of Zaragoza, Spain, for examination.

### 2.2. Macroscopic and Microscopic Examination of Fresh Tissues

Upon arrival at the laboratory, the samples underwent a selection process. We initially performed a macroscopic selection, followed by a microscopic evaluation to identify samples exhibiting macroscopic as well as microscopic cysts. In the end, we selected a total of 210 animals for this study.

We started by checking for macroscopic cysts on the surface, after which we made transversal cuts in the tissue portions with a scalpel. Microscopic cysts were visualized by applying the compression method: four separate fragments of each tissue (approximately 20 mm^2^ and 0.01 g) were firmly compressed and examined under optical microscopy at 50× and 100× magnification. Tissue fragments with one or more microscopic cysts were considered positive.

### 2.3. Freezing Treatment

Samples underwent a controlled homogeneous freezing process in a domestic freezer (3GF8601B, Aþþ, Balay, BSH Electrodomésticos España S.A., Zaragoza, Spain). They were kept at −20 °C for specific time intervals ranging from 24 to 144 h. Each sample consisted of 100–150 g of sheep meat. Samples were categorized into seven groups: fresh samples and samples frozen for 24 h, 48 h, 72 h, 96 h, 120 h, and 144 h. This process was prolonged until we obtained a total of 30 samples within each group. To calculate the sample size, we used WinEpi v2.0 (http://www.winepi.net/winepi2, accessed on 1 September 2019) with the estimated proportion option (random sampling and perfect diagnosis). To calculate an estimated proportion of 0% viable bradyzoites after the freezing procedure with an accepted error (or precision) of 10% and a confidence level of 95%, it was necessary to select an overall sample containing at least 27 individuals. The freezing chamber temperature and the temperature at the centre of the meat were measured with a temperature sensor (ALMEMO 2590-3S v5, Ahlborn, Holzkirchen, Germany) during the assay.

### 2.4. Pepsin Digestion

To facilitate the isolation of bradyzoites, we applied a pepsin digestion procedure using protocols recommended by Dubey (1998) and Bayarri et al. (2010) [35,36], with some modifications. Briefly, a 10 g portion of meat was crushed with 20 mL of NaCl (0.85%) with a mortar. Then, 20 mL of pepsin solution (pH = 1.1–1.2) was added, and the resulting mixture was incubated at 37 °C for 30 min with agitation. After digestion, the content was filtered and centrifuged for 10 min at 1500 g to separate bradyzoites. Then, 25 mL of NaHCO_3_ solution (1.2%, pH = 8.3) was added to neutralize the pepsin. Finally, after two PBS washes, the content was reconstituted in 2 mL of PBS. 

### 2.5. Molecular Identification

We performed molecular species identification on 14 samples we had randomly selected after successful isolation by sequencing. Both the *18S rRNA* gene and *Cox-1* gene were targeted according to Hoeve-Bakker et al. (2019) [37]. Genomic DNA was extracted from 200 µL of the resulting digestion using a commercial Speedtools DNA extraction kit (Biotools, B & M Labs, S.A., Madrid, Spain). Specific detection was carried out using a forward primer (SarcoFext: 5′-GGTGATTCATAGTAACCGAACG-3′) and a reverse primer (SarcoRext: 5′-GATTTCTCATAAGGTGCAGGAG-3′), which amplify a 900-bp fragment of the *18S rRNA* gene, designed by Moré et al. (2013) [38]. 

We developed a PCR test targeting an 1100 bp fragment of the *Cox-1* gene using previously described primers [39]: forward primer: 5′-ATGGCGTACAACAATCATAAAGAA-3′ (SF1) and reverse primer: 5′-ATA TCC ATA CCR CCA TTG CCC AT-3′ (SR9). The two PCR tests were performed in a 50 µL reaction mix containing 25 µL of HotStarTaq Master Mix, 40 pmol of each primer, 4 µL of template DNA, and RNase-free water. The touchdown PCR protocol for both the *18S rRNA* gene and the *Cox-1* gene included an initial 15 min. denaturation step at 95 °C, followed by 10 denaturation cycles at 94 °C for 30 s, annealing at 57 °C for 30 s with a decrement of 0.1 °C per cycle, and elongation at 72 °C for 90 s. The protocol continued with 35 cycles of 94 °C for 30 s, 56 °C for 30 s, 72 °C for 90 s, and a final extension for 10 min at 72 °C.

Sequence reactions (of a total volume of 20 µL) containing non-purified PCR product and 25 pmol of PCR primer according to company instructions were sent to STAB VIDA, Lda (Caparica, Portugal) for purified PCR amplicons and unidirectional Sanger sequencing. Obtained sequences were manually assembled and edited with the MEGA7 program [40]. A BLAST search (http://blast.ncbi.nlm.nih.gov/, accessed on 20 May 2020) was performed using the obtained nucleotide sequences for species classification. For BLAST analysis, >98% query coverage was applied to at least one of the sequenced genes. We deposited the generated *18S* and *Cox-1* sequences of *Sarcocystis* spp. from sheep in GenBank.

### 2.6. Evaluation of Bradyzoite Viability

After obtaining isolated bradyzoites through pepsin digestion, two vital stains were performed on each sample to assess their viability after freezing.

#### 2.6.1. Trypan Blue Staining

We applied this technique by following the protocol established by Strober (2015) [41]. The 0.4% dye was prepared in distilled water, and the ratio was established at 1:1. Briefly, 150 µL of a digestion resulting in 5 × 10^6^ bradyzoites/mL (Neubauer chamber) and 150 µL of 0.4% trypan blue were mixed in an Eppendorf tube and incubated for 3 min at room temperature. Five randomly chosen fields were observed under an optical microscope at 400× magnification. Stained (non-viable) and non-stained (viable) bradyzoites were counted during the next 3–5 min since bradyzoites die due to the reagent’s action (Figure 1).

#### 2.6.2. Double Fluorescence Staining

We based our approach on the sperm viability determination technique developed by Harrison and Vickers (1990) [42]. It applies two reagents: propidium iodide and carboxyfluorescein diacetate. We established the propidium iodide ratio at 1:50 and the carboxyfluorescein diacetate ratio at 1:100. This combination allowed us to differentiate between damaged (red fluorescent) and undamaged (green fluorescent) cells unequivocally. An excess of carboxyfluorescein diacetate causes the background of the preparation to acquire an excess of fluorescence, thereby preventing the identification of non-viable cells.

Briefly, 500 µL of a digestion resulting in 5 × 10^6^ bradyzoites/mL, 10 µL of carboxyfluorescein diacetate, and 5 µL of propidium iodide were inserted in an Eppendorf tube and incubated for 10 min at 37 °C. Then, 20 µL were deposited on a slide and observed under a microscope with a fluorescein filter of 450–490 nm excitation at 400× magnification with a long-pass filter. We then chose a total of 5 fields at random and counted the number of green (viable) and red (non-viable) fluorescent bradyzoites (Figure 2).

We calculated the viability percentage for each vital stain with the formula: Total viable cells/(Total viable cells + Total non-viable cells) × 100.

### 2.7. Statistical Analysis

Analysis was carried out using IBM SPSS Statistics for Windows, v24 (IBM Corporation, Armonk, NY, USA). Data normality was assessed using the Shapiro–Wilk and Kolmogorov–Smirnov tests. For non-normally distributed data, we used the non-parametric Kruskal–Wallis test to determine statistically significant differences among the different freezing times. Dunn’s post hoc test adjusted by Bonferroni was used for pairwise group comparison. We drew up a box and whiskers diagram to visualize viability according to freezing time and the staining method. Correlations between the two staining methods were determined using Spearman’s rank correlation (rs). Interpretation was performed according to Schober et al. (2018) [43], applying Spearman’s rank correlation, where rs = 0.00–0.10 = negligible correlation; 0.10–0.39 = weak correlation; 0.40–0.69 = moderate correlation; 0.70–0.89 = strong correlation and 0.90–1 = very strong correlation. Statistical significance was set at α = 0.05.

## 3. Results

### 3.1. Macroscopic and Microscopic Examination of Muscle

To carry out macroscopic examination and microscopic evaluation of the selected samples, we used the compression method. Macroscopic cysts in animal muscle are shown in Figure 3, and microscopic cysts are shown in Figure 4.

### 3.2. Molecular Analysis

The 14 selected samples generated sequences with suitable sizes that could be targeted for species classification (Table 1). Considering the 98% cut-off value for the *18S* gene, we found *S. tenella* in three animals, *S. medusiformis* in three animals, and *S. gigantea* in four animals. According to *Cox-1* gene sequencing, *S. tenella* was detected in three animals, *S. medusiformis* in two animals, and *S. gigantea* in six animals.

### 3.3. Viability Study after Freezing

Viability percentages decreased as a function of the number of hours a sample was frozen. They were similar among the two staining methods: 92–93% parasite viability in fresh samples, 33% after 24 h of freezing, 12–13% after 48 h, 6–8% after 72 h, 1.5% after 96 h, 0–1.5% after 120 h, and 0% after 144 h of freezing. Statistical analyses revealed significant differences in terms of viability among the various treatments (*p* < 0.001 for each of the two techniques). After 72 h of freezing, the decrease became less pronounced, with similar viabilities at 96, 120, and 144 h of freezing (viability in the case of both techniques was consistently less than 10%, with median values ranging from 1.5% to 0%). The viability decrease between fresh samples and those frozen for 48, 72, 96, 120, and 144 h was statistically significant. Similarly, viability decreased significantly when comparing samples frozen for 24 h with those frozen for 96, 120, and 144 h (Table 2, Figure 5). 

Spearman’s correlation test showed a significant and very strong correlation between the two techniques (rs = 0.971; *p* < 0.001). The graphic representation of correlation in these data indicates a linear tendency (Figure 6).

## 4. Discussion

After molecular examination, our research identified three species of *Sarcocystis*: *S. tenella*, *S. gigantea*, and *S. medusiformis*. Interestingly, we did not detect *S. arieticanis* despite its worldwide distribution in tandem with *S. tenella* and *S. gigantea*. On the other hand, we found a less common species, *S. medusiformis*. This species is mainly restricted to Australia and New Zealand, with two isolated descriptions in Iran [44] and Italy [45]. However, a recent study by Gjerde et al., 2020 [8], confirmed its presence in Spain, and we concur with their hypothesis that the parasite has been present in European domestic sheep populations for a long time but had not been previously detected. Cases of co-infection of multiple *Sarcocystis* species in the same animal cannot be ruled out: previous studies [46,47] have observed a co-infection of *S. tenella* and *S. arieticanis* in sheep. As Sanger sequencing only allows for the detection of a single sequence, we probably only detected the dominant species in each animal. Selecting a single microcyst or macrocyst for Sanger sequencing should help address this issue.

Regarding temperature and minimum freezing treatment time for inactivation of bradyzoites, we observed a significant decrease in viability (12–13%) after the first 48 h of freezing. This loss of viability remained significant until 96 and 144 h, when viability was 1.5 and 0%, respectively. Our results differ from those obtained by Koudela and Steinhauser (1984) [10], who assessed viability using DAPI fluorescent staining on *Sarcocystis* spp. in beef meat. Using biological assays in definitive hosts, they ascertained non-survival of the parasite at −18 °C, 48 h.

Collins and Charleston (1980) [17] investigated the viability of *S. gigantea* after freezing; they found that cysts remained infectious for 60 days following treatment at −14 °C and lost infectivity after 516 days. Those results differ significantly from our study, where hardly any viability could be observed after 96 h at −20 °C. These discrepancies may arise from the cysts’ varying resistance to −14 °C compared to −20 °C as well as from methodological differences. Those authors assessed infectivity by feeding cats with cysts and monitoring the presence of oocysts or sporocysts in their faeces; moreover, they did not evaluate viability in the period between 60 and 516 days.

Chen et al. (2007) [48] examined the effects of −20 °C freezing over extended periods on *Sarcocystis* structure. They found that bradyzoites degenerate and dissolve with freezing, resulting in a probable loss of bradyzoite activity and suggesting that cysts lose their infective capacity. To our knowledge, ours is the first study to specifically assess viability in sheep meat after freezing at −20 °C; nevertheless, the effect of the same temperature on intermediate hosts harbouring other *Sarcocystis* species has been examined in previous studies. For example, freezing at −20 °C for 24–48 h is sufficient to inactivate *S. suihominis* and *S. miescheriana* in pig [9], *S. levinei* in buffalo [12], *S. capracanis* in goat [13], and *S. fayeri* in horse [14]. However, one specific study differs from these findings and ours: only 2 h at −20 °C were required for *S. sybillensis* and *S. wapiti* inactivation in Sika deer (*Cervus nippon centralis*) meat [16]. In domestic freezing, the effect rate is slow. Freezing leads to the formation of large, pointy ice crystals capable of disrupting cell membranes; the concomitant loss of membrane integrity results in parasite death [48]. In theory, longer freezing times lead to the growth of more crystals and, thereby, to a higher percentage of cells with damaged membranes. Our results suggest that the freezing of meat for 96 to 144 h is effective for inactivating the parasite: after those time periods, median viability drops to 1.5 and 0%, respectively.

Regarding the vital stains applied in our study, trypan blue staining has been widely employed for counting viable cells using optical microscopy without fluorescence [46]. In terms of cost, setup complexity, and the time required for implementation, we concur with Strober (2015) [41] that trypan blue staining is a straightforward, rapid, and cost-effective method for the viability assessment of bradyzoites. 

Regarding carboxyfluorescein diacetate and propidium iodide, double fluorescence staining has not been previously applied to *Sarcocystis* spp. bradyzoites to the best of our knowledge. However, this technique is more expensive and laborious, as it requires an incubation period with the additional cost of an incubator [42] and a fluorescence microscope. However, when it comes to interpreting results, this technique is much more convenient, as green-stained viable bradyzoites and red-stained non-viable bradyzoites can be distinguished unequivocally. Flow cytometry is commonly employed in parasite viability studies for several different purposes [49,50], and it has proven to be more useful than fluorescence microscopy in those contexts. Whereas fluorescence microscopy is effective for the quantification of cellular components in quantities of dozens or hundreds of cells, flow cytometry excels in the analysis of larger quantities, reaching up to hundreds of thousands of cells in suspension. Thus, flow cytometry is well-suited for rapid quantitative analysis of a considerable number of cells, while fluorescence microscopy provides more detailed morphological observation of cells within their tissue context.

With these staining techniques, we can demonstrate whether bradyzoites are viable or not, but we cannot demonstrate infectivity. It would thus be valuable to investigate the parasite’s infective capacity even in samples with low viability. It would be interesting for future studies to assess the actual infective capacity of the low percentage of bradyzoites that remain viable after freezing. An extrapolation of those staining methods to zoonotic species, particularly pigs, would undoubtedly show promise and merits further investigation.

Finally, our results confirm that sheep meat consumption can represent a risk to human health. It is crucial for farmers to be trained to refrain from feeding their dogs and cats with raw meat in order to effectively disrupt the infection cycle between the intermediate and definitive hosts [51].

## 5. Conclusions

Molecular analysis confirms the presence of *S. tenella*, *S. gigantea*, and *S. medusiformis* in domestic sheep populations in Spain. Furthermore, viability amounted to 33% after 24 h of freezing, decreasing to 12–13% after 48 h. A significant reduction in bradyzoite viability was noted when muscular tissues were subjected to freezing at −20 °C for a total of 96 h. Additionally, we observed a strong correlation between the trypan blue and double fluorescence staining methods. Both are thus effective in assessing the viability of *Sarcocystis* spp., and their results can be applied to further species that have zoonotic implications.

## Figures and Tables

**Figure 1 vetsci-11-00103-f001:**
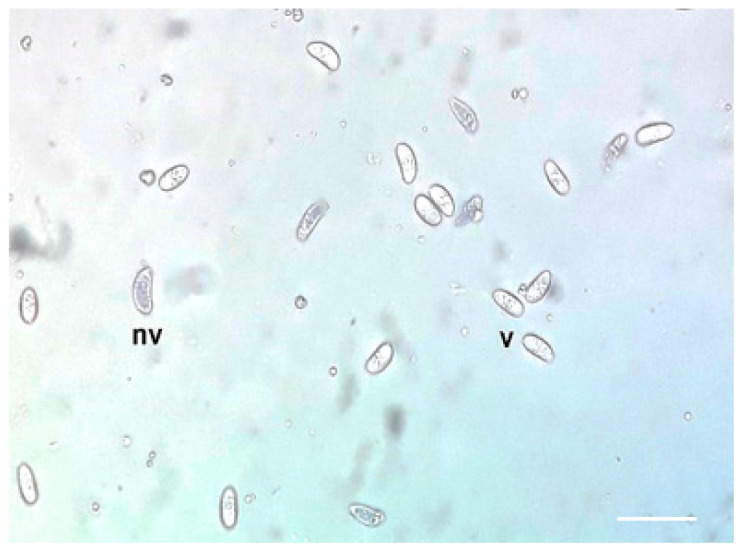
Trypan blue stain of *Sarcocystis* bradyzoites. Viable bradyzoites (v) and non-viable bradyzoites (nv), 400× magnification.

**Figure 2 vetsci-11-00103-f002:**
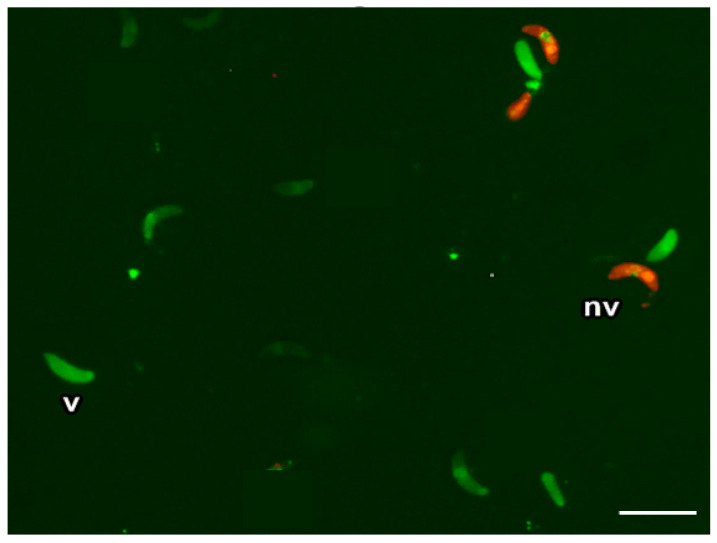
Double fluorescence staining of *Sarcocystis* bradyzoites. Viable bradyzoites, green (v), and non-viable bradyzoites, red (nv), 400× magnification.

**Figure 3 vetsci-11-00103-f003:**
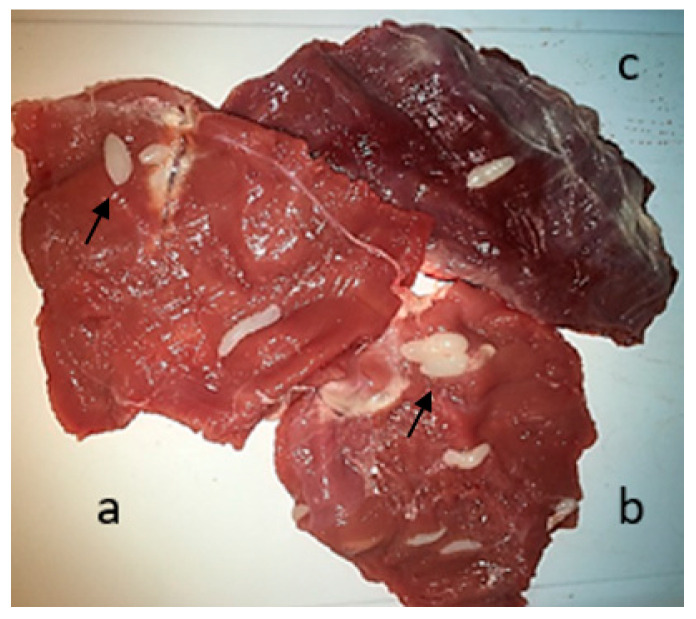
Gross appearance of *Sarcocystis* macrocysts (indicated by arrows) in the hind leg (**a**), foreleg (**b**), and abdominal muscles (**c**) of slaughtered sheep.

**Figure 4 vetsci-11-00103-f004:**
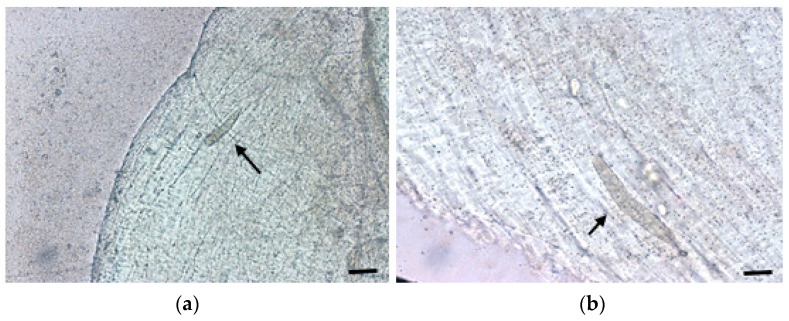
Light microscopic appearance of a *Sarcocystis* microcyst (indicated by arrows). (**a**) 50× magnification, (**b**) 100× magnification.

**Figure 5 vetsci-11-00103-f005:**
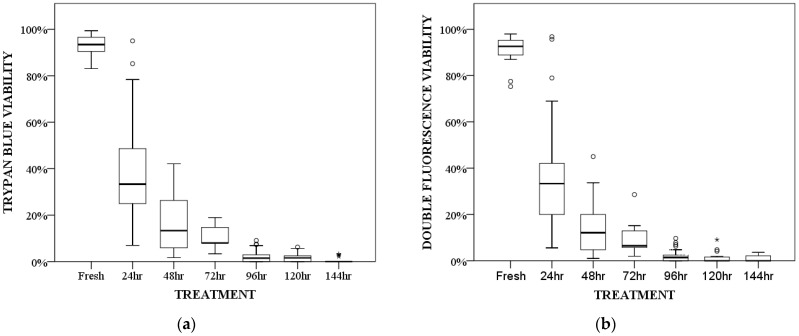
Box plot representation of viability after different freezing treatments for the two staining techniques. (**a**) Trypan blue and (**b**) double fluorescence. Data are presented as boxes and whiskers. Each box includes the 25 and 75 interquartile; the line inside the box represents the median, and the whiskers represent the minimum and maximum values. Outliers are indicated by a circle, and extreme outliers by asterisks.

**Figure 6 vetsci-11-00103-f006:**
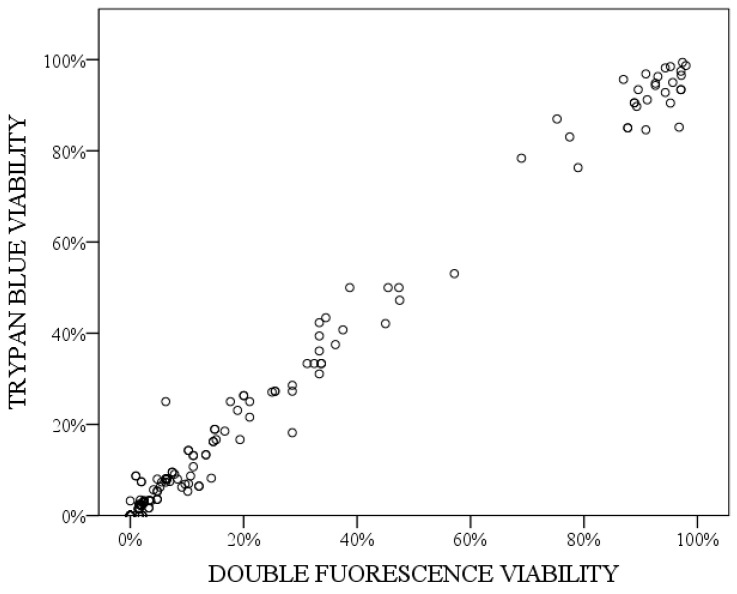
Graphic representation of Spearman’s correlation between trypan blue staining and double fluorescence staining.

**Table 1 vetsci-11-00103-t001:** *Sarcocystis* spp. results after *18S* and *Cox-1* gene sequences.

*18S* Gene				*Cox*-*1* Gene			
*Sarcocystis* spp.	%	ID	Length	*Sarcocystis* spp.	%	ID	Length
*S. tenella*	100%	KP263759.1	1804	*S. tenella*	99%	MK420001.1	1038
*S. medusiformis*	100%	MK420021.1	1928	*S. medusiformis*	92%	MK420014.1	1038
*S. gigantea*	99%	MK420020.1	1910	*S. gigantea*	100%	MK420011.1	1038
*S. gigantea*	98%	MK420020.1	1910	*S. gigantea*	99%	MK420013.1	1038
NV	NV	NV	NV	*S. gigantea*	99%	MK420013.1	1038
*S. tenella*	98%	MF401626.1	837	*S. tenella*	99%	MK419987.1	1038
*S. gigantea*	98%	MK420020.1	1910	*S. gigantea*	(99%)	MK420011.1	1038
*S. tenella*	99%	KP263759.1	1804	*S. tenella*	(99%)	MK420001.1	1038
*S. gigantea*	90%	MK420020.1	1910	*S. gigantea*	(99%)	MK420011.1	1038
NV	NV	NV	NV	*S. medusiformis*	99%	MK420014.1	1038
*S. gigantea*	99%	MK420020.1	1910	*S. gigantea*	99%	MK420011.1	1038
*S. medusiformis*	100%	MK420021.1	1928	NV	NV	NV	NV
*S. medusiformis*	99%	MK420020.1	1910	*S. medusiformis*	90%	MK420014.1	1038
*S. medusiformis*	87%	MK420021.1	1928	*S. medusiformis*	99%	MK420015.1	1038

NV = invalid sample; % = percentage sequence homology; ID = sequence identification.

**Table 2 vetsci-11-00103-t002:** Percentage of viability (%) for the two staining techniques and significant differences among different lengths of freezing time. The results are expressed as the median, minimum, and maximum.

Hours at −20 °C	Trypan Blue ^1^	Double Fluorescence ^1^
Fresh	93.42 (83.05–99.38) ^a^	92.59 (75.25–97.96) ^a^
24	33.33 (6.98–95) ^a,b,c^	33.33 (5.56–96.77) ^a,b,c^
48	13.33 (1.67–42.11) ^b,c^	12.12 (0.99–45) ^b,c^
72	8 (3.33–18.92) ^c^	6.45 (1.96–28.57) ^c^
96	1.47 (0–9.09) ^d^	1.42 (0–9.68) ^d^
120	1.58 (0–6.25) ^d^	0 (0–9.09) ^d^
144	0 (0–3.23) ^d^	0 (0–3.61) ^d^

^1^ Superscripts with different letters in a column indicate significant differences (*p* < 0.05).

## Data Availability

The data supporting this study’s findings are available on request from the corresponding author, [M.-P.P.]. Sequences obtained in this study are deposited in GenBank with the accession numbers: PP227426-PP227429, PP227418-PP227420, PP218653-PP218655, 2796007, 2796002, and 2795997.

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
