# Peer review of "Identification of Sarcocystis spp. in Slaughtered Sheep from Spain and Evaluation of Bradyzoite Viability after Freezing"

_vetsci, 2024, doi:10.3390/vetsci11030103_

Round 1

Reviewer 1 Report

Comments and Suggestions for Authors

Early detection of parasite infections is very crucial in the food industry. Scientists do an excellent job of testing different temperature conditions, and the impact on viability of the parasite.

It is very interesting to see the use of Trypan Blue, and combined fluorescent stains for parasite detection and viability.

Here are my concerns in the manuscript.

1. Authors talk about different heat treatments in Figure 5 however, it is not mentioned in the methods section. It is not clear if the plots represent two different parasite strains or heat treatment conditions.

2. As the fluorescent stain is extensively used in Flow Cytometry analysis, the authors should have compared the accuracy of data for fluorescence microscopy vs FACS.

3. With the microscopy advancing rapidly, the authors should have provided DIC images showing the borders of the parasite and then merged versions. The red and green stain could be overlapping debris and cells. But it is not evident that they are colocalized. 

4. Figure 2 is inconclusive. The resolution of the figures should be improved.

5. The figures 1 and 2 are missing the scale.

Reviewer 2 Report

Comments and Suggestions for Authors

Manuscript ID: vetsci-2820312

Title: Sarcocystis spp. identification in slaughtered sheep from Spain and evaluation of the bradyzoite viability after freezing treatments

Authors: María Paz Peris *, María Jesús Gracia, Bernardino Moreno, Paula Juan Puente, Mariano Morales, María Serrano, María Dolores Manzano, Nabil Halaihen, Juan Badiola, Juan Antonio Castillo

 General Comments:

If accepted, this manuscript will require a moderate amount of grammatical editing, though these errors do not interfere with the meanings of the text. The manuscript reports the finding of two unrelated objectives that use the same tissue. The first objective consists of a very standard molecular survey of micro and macro sarcocysts harvested from sheep in central Spain. This survey used established PCR and Sanger sequencing methods involving both the 18S rRNA gene and Cox-1 gene. For the second objective, the authors used sarcocyst-infected tissues to evaluate the effects of freezing on the viability of bradyzoites isolated from the tissue. Viability was assessed in vitro using two staining methods: trypan blue and a double fluorescence stain involving propidium iodide and carboxi-fluorescein diacetate. In its current form, the molecular study provides no new information on the presence of Sarcocystis species in Spain because a similar recent study from Spain found similar results based on a larger sample size. The freezing study was able to show a significant diminishment in in vitro viability with time, however, they did not determine which species of Sarcocystis they were evaluating. This is a serious omission that needs to be corrected.

Title:

1.     The title is appropriate.

Abstract and Simple Summary:

1.     The location of the study is not specifically mentioned at the beginning of the abstract. It is implied later with the sentence, “The study conclusively confirmed the presence of S. tenella, S. gigantea, and S. medusiformis in Spanish domestic sheep”, however, the location should be specifically stated.

2.     There are missing words and punctuation in the abstract. These need to be corrected.

3.     The first sentence in the simple summary is misleading. The study did not “advance our understanding of identifying Sarcocystis spp. in Spain”, but rather identified the species using a previously described molecular method. The method for inactivating the bradyzoites is not innovative and multiple strategies were not evaluated. Freezing at -20C was the only method tested and this has been used for various parasites (including Sarcocystis) for years.

Keywords:

1.     I don’t understand the phrase “domestic freezing”. I think that there is a mistake with this keyword.

Introduction:

1.     Didn’t Gjerde et al. (2020) identify 5 species in sheep from Spain? Why does lines  38-40 list only 4?

2.     Wasn’t S. medusiformis identified in Italy? I think that Italy is considered part of southern Europe.

3.     The authors imply that it is important to evaluating the effects of freezing sheep meat to diminish the infectivity of Sarcocystis species, however, the potential differences in the effectiveness of freezing exists in the different susceptibilities of the species, not in differences in the source of meat. This should be clarified.

Methods:

1.     A total of 210 when the sample were selected for the study, but the text suggests that only tissues with both macroscopic and microscopic cysts were selected. If this is not true, then lines 115-117 should be adjusted to clarify this. If it is true, why was this approach taken. Is it possible that this might cause a bias in the findings.

2.     The biggest problem with the method section is the lack of information about the number of sheep tested. A total of 210 cysts were selected, but how many animals did these come from. If it was from only 3 sheep, then this would provide a different perspective than if was from 30. Were these sheep representative of the typical one in the area or  did they provide a source of bias. Were older ewes specifically selected or did this include young lambs. This host information must be provided.

3.     Only 14 of the 210 samples were used for the molecular study. These 14 samples were randomly selected; therefore, it is possible that some of these samples cluster from within only a few sheep. It would have been better if the cysts were specifically selected from different sheep.

4.     The length of time needed to freeze tissue is dependent on the size/weigh of the sample. The weight of the tissue prior to freezing should be provided.

5.     Is it possible to know which species were being tested for viability? If so, then this should be reported.

6.     Why was the trypan blue diluted into distilled water? When equal amounts of PBS containing bradyzoites was mixed with the trypan blue, the hypotonic solution may have negatively affected the bradyzoites.

7.     For the double fluorescence stain, a 450-490 nm excitation filter was used, but what wavelength emission filter was used. Given that both red and red was seen in figure 2, I’m assuming that it was a long-pass filter, but this should be stated.

Results and Figures:

1.     I’m not sure that figures 1-4 are needed for the manuscript. If they are included, they need to include scale bars instead of listing magnifications in the figure caption.

2.     The section entitled, “Macroscopic and microscopic examination of muscle”, provides not actual results and should be deleted from this section.  

3.     Figure 1 contains 2 photos, but they both appear to include viable and non-viable bradyzoites. Why are both included (i.e. how are they different?). Why are the background colors so different? It seems like the right-hand photo can be eliminated.

4.     Figure 3 should include labels and arrows to point out a couple of macrocysts.

5.     Figure 4 should include a label for the microcyst. Why is the low magnification photo needed?

6.     Image quality for Figure 5 is very poor. By adding the information in Table 1 to the graphs in Figure 5, you could delete the table.

7.     The results from the species identification study provides no new findings. Verifying the presence of S. medusiformis in 4 of the 14 samples has some value, but Gjerde et al. (2020) already reported it in 11 of the 63 samples that they tested from the same general area.

8.     The value of the freezing study is diminished because the authors did not determine the species of Sarcocystis being tested. The species of Sarcocystis would likely affect it’s tolerance to freezing more than the species of host that it’s in. It is possible that each species has a different tolerance to freezing, and yet there is no way to determine this based on the approach used in this study,

9.     The comparison of the 2 staining techniques was good.

Discussion:

1.     The discussion of the molecular results is very brief and this reflect the low significance of these results in their current form. If the identities of the sheep are know so that more epidemiological analyses can be made, then this would add value to this section. Were the genotype identified in this study similar to those seen by Gjerde et al. (2020)? A more extensive analysis of the molecular data might also add some additional value.  

2.     The discussion should explain the difference between viability and infectivity. If this study is evaluating the value freezing meat to eliminating infections in the definitive host, what are the risk of using in vitro viability as the criteria? In lines 307-312, the authors state that this study “suggest that freezing meat for 96 to 144 hours is effective in inactivating the parasite”. This statement should be qualified in the context of viability versus infectivity.

3.     The discussion should also address the issue of not knowing the species of Sarcocystis used for the freezing study. Are there reasons why knowing the species is not important?  

4.     The authors state that, “Van Bree et 304 al. (2018) [45] indicate that there is no risk of transmission to animals when raw meat is 305 frozen at -20 ËšC for one or two days for S. tenella and S. cruzi”, however, this 2018 study appeared (based on the abstract) to only evaluated the presence or absence of these 2 Sarcocystis species in frozen samples. It does not appear to include any evaluation of their infectivity. They may have speculated on the infectivity, but this should be clarified in the current discussion.

Conclusion: 

1.     The conclusion is an appropriate summary of the findings for this study.   

References:

1.     The authors need to check the appropriateness of their references. For example in line 78, they list 3 references (i.e. 14, 24 and 25) in a sentence that seems to deal with the use of trypan blue for evaluating the viability of bradyzoites from different Sarcocystis species; while reference 14 does, 24 and 25 do not.

Comments on the Quality of English Language

Reviewer 3 Report

Comments and Suggestions for Authors

The study is relevant, but major corrections are needed. Methodological description must be improved, the choose of samples broadly explained. Figures and their description must be improved. Description of the results must be improved. DNA sequences obtained must be submitted to GenBank. Explanation on how you got pure DNA sequences must be given taking into account that Sarcocystis spp. co-infection are very common in sheep.

L1 Sarcocystis in italic should be

L11-17 please expand simple summary, describe in simple terms what Sarcocystis parasites are and how they are important in veterinary medicine

L24 dot between sentences is missing

L38-41 data is not correct, four well described species and two rarely found species S. microps and S. mihoensis, please include

L47 “carcasses affected by two or more muscular areas” affected by what? it is missing macrocysts or..

L53-54 or L38-41 authors must point out that Sarcocystis spp. producing macrocysts in sheep (S. gigantea and S. medusifomis are transmitted via cats, while S. arieticanis and S. tenella which rarely can induce acute infections are transmitted via dogs.

L55 text not intended

L57 suihominis

L57-59 not all listed species are zoonotic

L55-67 some conclusions are needed that viability of Sarcocsytis spp. after  exposure to freezing depends largely on the species of the parasite

L76 Sarcocystis italic

L78 S. sinensis is not specific for cattle. This species is specific for buffaloes. This must be corrected

L80 “interesting staining method” correct. why interesting, for what procedures is it applicable

L92-93 it is not correct, Gjerde et al. 2020, they examined Sarcocystis spp. in sheep from Spain, and this work was cited by authors of this study.

L113-136 it is not clear according to what criteria did you select 30 (L133) out of 210 (L117) samples for the freezing analysis?

155-161 include the names of the primers for both genes, this will make it easier for those who try to repeat the procedure you suggest.

Table 1 should appear immaterially after you mentioned it in the text, but in your case, you have cited table 1 at L171 and it appear at L251. Please correct it. I think here L171 you shouldn’t cite Table 1

L172 why unidirectional sequencing was chosen? In such case you lose some information at the 5’ end, also there is no repetition for the validity of obtained results

L100 and further. According to MDPI style, Sub-section heading is written in such style that first letters are capitalised

L182 correct style

L186 for me this methodological peculiarly is not clear. How do you known what is bradyzoite concentration?  “5 x 106 bradyzoites/mL concentration” shouldn't the concentration depend on the parasite load/infection intensity?

L192-194 correct description of the figure legend; no two images are provided and it is not explained what are the differences between them (a,b). the scale bar is missing

L210-216 please include in the text and figure caption: viable (green), non-viable (red)

L218-219 how you managed to calculate all bradyzoites, for instance from Figure 2a? it seems to be to challenging due to high concentration of bradyzoites

L236-237 English must be improved

Figure 3 indicate in the figure different muscle types

Figure 4 must be improved. use the arrows to show where the cyst is. Figure 4a resolution must be improved or other image used. Magnification is not 5× and 10×, it is 50× and 100×. Scale bar is missing

3.1 the description is too short. What was prevalence of macrocysts and microcysts?  L115-117 not clear how many samples had macrocysts and how many microcysts. What was differences of the prevalence in different muscle types. It is not directly related to your study on the freezing, but the prevalence data is very important for the general comparison, also data of macrocyst infection rates in sheep are scarce and in some parts of Europe very low.

3.2 I do not understand your idea on molecular results, why they this was performed? Why just few sarcocysts were analyzed, how they chosen from which types of muscles, in how many of them microscyts and macocysts were visible, both macrocysts and microcysts or just eitgher microcysts and macrocyst? (information on whether you found microcysts and/or macrocysts in samples you have sequenced should be included in Table 1). Your sequences are not available in GenBank, so you have no proofs of your results. Now without GenBank accession numbers, it can be said that molecular analyses are not performed at all. Table 1 capture “NV = not valid sample” there is no explanation why not valid. In sheep co-infection with several species of Sarcocystis are very common, and most of your samples should be infected with S. tenella and S. arieticanis as it shows various recent works (you have not cited them…). My concern is regarding your sequencing results. You have extracted DNA from digestion isolates, which should contain in most cases 2-3 species (if samples were infected with both microcysts and macrocysts). When you used Sarcocystis species primers which should target all species analysed, so the following question and concern is on the quality of your sequences, whether there were double peaks, or multipeaks? How can you explain that only one species was amplified, is this species dominant in given sample?

Figure 5 resolution power is too low, names are almost impossible to read

L251 in the table it should be “Sarcocystis spp.”

L254-262 the description of data must be improved. Please provide more values in the text. In fresh median more than 90% of viability, after 24 h 33%, after 28h 12-13%. It should be mentioned that very similar results were obtained by both dyeing methods.

Figure 6 the graph is good but please use a larger font for viability methods and for percentage

L282 Gjerde

L277-284 the explanation of why S. arieticanis was not found should be included. Please see remarks 3.2… my suggestion would be that whole amount of S. ariecanis is lowest in samples (due to low parasite load of this species, maybe?). It also should be discussed about mix-infection of Sarcocystis spp. when in single sheep sample  several different Sarcocystis species are present.

L336-337 please add values; I suggest top add values of viability after 24 and 48 hours it is very important results.

L287 include values of viability

L319-326 include references

Discussion on viability of Sarcocystis in different hosts should be broadened, you should compare your results with data presented in L57-67 and L76-79. Please take in mind that in your case it were at least 3 species, but most likely four species (plus S. arieticanis) this fact should be included in discussion on viability.

The manuscript lacks important references on Sarcocystis spp. in sheep:

Doi: 10.1007/s11250-015-0789-4 

Doi: 10.1007/s12639-014-0495-6 

Doi: 10.1051/parasite/2017025

Doi: 10.3390/ani9050256

Doi: 10.1007/s00436-020-07002-w

Doi: 10.3390/ani12162048

Doi: 10.3390/vetsci10080520

Doi: 10.1016/j.cimid.2021.101738

Doi: 10.3389/fvets.2023.1225796

Doi: 10.3390/pathogens12070902

Doi: 10.1007/s00436-022-07469-9

Recommend adding the above references.

L342-349 this is not according journal style reccomendations, abbreviation of authors must be used

L357-358 GenBank numbers!

References: please use italics for species, genus names. Dots are absent or included at the end of reference, please corrcect. Vet. Parasitol. Should be. There is no abbreviation of “Infection and Immunity”. correct [26] journal name in italic. etc…

Comments on the Quality of English Language

L236-237 English must be improved

Round 2

Reviewer 2 Report

Comments and Suggestions for Authors

The authors have corrected many of the suggestions that I made, and these changes were appropriate and added value to the manuscript. The authors disagreed with a few of my suggestions or it wasn't possible to add the requested information in a couple of cases. While I do believe that these suggestions would have added value to the manuscript, they were not significant enough to interfere with the acceptance of this manuscript based upon the responses that they gave. 

Author Response

Indeed, your suggestions have greatly improved the article. For this I thank you on behalf of all the authors of the article.

Reviewer 3 Report

Comments and Suggestions for Authors

Thank you for your answers and corrections, below are comment what should be addressed.

L13 maybe change to “One of the most significant”, since this does not undermine the results on viability, which in my opinion is more important than the identification of S. medusiformis

L19 use apicomplexan instead of protozoan

L50 Sarcocystis spp.

L63 Sarcocystis

L85 Latin name of host species is needed to be presented in brackets

L143 improve English wording “we should select”…maybe .Based on …. analysis it was suggested to examine the sample with at least 27 individuals…

L191 thank you for providing your sequences in GenBank, now your data can be checked and approved. However, please correct your records. Organism=Sarcocystis, not sheep host=sheep

L257-260 please include how many of 27 analysed sampled for viability analysis had i) microcysts ii) micorcysts and macrocysts. This should be clearly written in the manuscript.

L267-268 in the figure caption must be written, arrows points at sarcocysts.

L270-276 in the Table results of both genes 18S rRNA and cox1 are presented, but why only 18S rRNA sequences are submitted to the GenBank, why you haven’t submitted cox1 sequences? Based on PP218653- PP218655 sequences, S. tenella cannot be distinguished from S. capracanis; therefore, cox1 is essential.

L401-402 here GenBank accession numbers must be included

Figure 5 font size must be increased, I can not read what is written. Authors ignored this remark in previous round of review.

L335 capacity.

Comments on the Quality of English Language

English needs minor editing, for example L143 improve English wording “we should select”…(maybe it can be changed to..) Based on …. analysis it was suggested to examine the sample with at least 27 individuals…
